# *MiR-532-3p* inhibited the methylation of *SOCS2* to suppress the progression of PC by targeting *DNMT3A*

Kaiqiong Wang*, Dongwei Gong*, Xin Qiao, Jinfang Zheng

**Pancreatic cancer (PC) is one of the deadliest malignancies, with poor diagnosis and prognosis. *miR-532-3p* has been reported to be a tumor suppressor in various cancers, whereas the mechanism of *miR-532-3p* in the progression of PC remains poorly understood. In this study, it was found that *miR-532-3p* and *SOCS2* were down-regulated, whereas *DNMT3A* was up-regulated in PC. Knockdown of *DNMT3A* or overexpression of *miR-532-3p* suppressed PC cell proliferation, invasion, and migration, as well as tumor formation in nude mice. DNMT3A induced the methylation of *SOCS2* promoter. *SOCS2* knockdown reversed the inhibiting effect of *DNMT3A* silencing on PC cell growth. *miR-532-3p* directly bound to *DNMT3A* and negatively regulated its expression while up-regulating *SOCS2* levels. *DNMT3A* overexpression reversed the inhibiting effect of *miR-532-3p* overexpression on PC cell growth. In conclusion, the overexpression of *miR-532-3p* could suppress proliferation, invasion, and migration of PC cells, as well as tumor formation in nude mice through inhibiting the methylation of *SOCS2* by targeting *DNMT3A*.**

## Introduction

Pancreatic cancer (PC) is the third leading cause of cancer-related death in the United States and the fourth leading cause in Europe, facing huge challenges in diagnosis, treatment, and prognosis (Ferlay et al, 2018; Siegel et al, 2021). Although gemcitabine has been demonstrated to be effective in improving the median survival time of patients with PC, the response rate is only 29% for first-line therapy with albumin-bound paclitaxel plus gemcitabine and the prognostic outlook is unsatisfactory (Hu et al, 2022). According to statistics, the 5-yr survival rate of patients with PC is about 3%, and the 10-yr overall survival rate is less than 1%, which is the lowest survival rate of all malignant tumors (Siegel et al, 2018). Thus, it is urgent to investigate the pathogenesis of PC, to identify novel biomarkers with high specificity, and to develop more effective targeted therapeutic strategies.

DNA hypermethylation and hypomethylation are key factors in tumor development by inhibiting tumor suppressor gene expression and inducing genomic instability (Jones & Baylin, 2002; Karpf & Matsui, 2005), which are catalyzed by kinds of DNA methyltransferases (DNMTs), such as DNMT1, DNMT3A, and DNMT3B, via transferring a methyl group to the 5′ position of cytosine in the CpG island (Robertson, 2001). The abnormalities of DNMTs have been linked to the progression of PC in the study of tumors. *DNMT3B* was reported to be overexpressed in PC tissues and cells, and as a targeted gene of *miR-29b*, *DNMT3B* silencing contributed to the apoptosis of PC cells (Wang et al, 2018). *DNMT3A* had also been shown to be highly expressed in PC and was involved in the proliferation and cell cycle progression of PC cells (Jing et al, 2019). Therefore, DNMTs may be crucial genes that have the potential to drive the development of PC treatments. Suppressor of cytokine signaling 2 (*SOCS2*) is one of the key proteins that regulates cytokine responses and has been reported to be down-regulated in tumors, such as breast cancer and ovarian cancer (Sutherland et al, 2004; Slattery et al, 2014). Abundant CpG islands were found to exist in the promoter region of *SOCS2* through bioinformatics analysis, and a previous study indicated that *SOCS2* CpG islands showed hypermethylation in endometrial cancer (Fiegl et al, 2004). Furthermore, in the study of colon cancer, Xu et al suggested that methyltransferase-like 3 regulated the levels of *SOCS2* through modulating methylation-mediated *SOCS2* RNA degradation, thereby disrupting the proliferative ability of tumor cells (Xu et al, 2020), indicating that the changes of *SOCS2* expression may be related to the methylation modification. However, the methylation status of *SOCS2* at the DNA level and the role of *SOCS2* in PC remain unclear. The data from The Cancer Genome Atlas (TCGA; http://www.cbioportal.org) demonstrated that *SOCS2* was down-regulated in PC. We therefore speculated that DNMT3A might be involved in regulating the methylation level of *SOCS2*, which requires further investigation.

miRNAs are 19–24 nucleotide bases of noncoding RNA and have been reported to be involved in post-transcriptional regulation of gene expression (Friedman et al, 2009). Growing evidence suggested that miRNAs were epigenetically silenced in several human malignancies and that their dysregulation was involved in the

---

Department of Hepatobiliary Surgery, Hainan General Hospital, Haikou, P.R. China

Correspondence: zhengjinfang0902@163.com
*Kaiqiong Wang and Dongwei Gong are co-first authors.

---

 

proliferation, apoptosis, and metastasis of tumor cells (Suzuki et al, 2012). *miR-532-3p* has been extensively studied as a tumor suppressor gene in a variety of cancers, such as lung cancer (Jiang et al, 2019) and colorectal cancer (Gu et al, 2019). Notably, the *miR-532-3p* expression has been reported to be controlled by DNMT3A-mediated methylation in the promoter region of its host gene (Zhou et al, 2018). The role of *miR-532-3p* in PC has not been reported yet; however, TCGA data indicated a low expression of *miR-532-3p* in PC, and StarBase (http://mirwalk.umm.uni-heidelberg.de) predicted a binding site between *miR-532-3p* and *DNMT3A*. Thus, exploring their regulatory relationship and related pathways may provide more valuable insights into the treatment of PC.

In this study, we discovered that *DNMT3A* was up-regulated in PC and regulated the *SOCS2* expression. Importantly, we discovered a binding site between *DNMT3A* and *miR-532-3p*. Our findings illustrated the role of the *miR-532-3p/DNMT3A/SOCS2* pathway in inhibiting the progression of PC, providing a potential target for disease improvement.

# Results

### *DNMT3A* was aberrantly up-regulated in PC tissues and cells

*DNMT3A* had been reported to be overexpressed in PC, and we further confirmed this conclusion. The results showed that, compared with normal pancreatic tissues, DNMT3A was highly expressed in PC tissues (Fig 1A and B). Furthermore, the association between *DNMT3A* overexpression and the clinicopathological status of PC patients was investigated. TNM was significantly associated with higher *DNMT3A* expression, as shown in Table 1. Similarly, the data from in vitro experiments also showed that the mRNA and protein levels of *DNMT3A* in PC cells were higher than those in human pancreatic duct epithelial (HPDE6-C7) cells (Fig 1C and D). The differential expression in PANC-1 and Capan-1 cell lines was the most obvious, which were selected for the subsequent study.

### *DNMT3A* knockdown reduced the proliferation, migration, and invasion of PC cells, as well as tumor formation in nude mice

The knockdown efficiency of sh-*DNMT3A* in PANC-1 and Capan-1 cells was determined by assessing the DNMT3A expression at mRNA and protein levels. As shown in Fig 2A and B, the DNMT3A expression was significantly reduced after transfecting with sh-*DNMT3A* in vitro, providing an effective transfection for subsequent experiments. Next, the proliferative, migration, and invasion abilities of PANC-1 and Capan-1 cells were analyzed. The results indicated that the knockdown of *DNMT3A* significantly inhibited PC cell proliferation (Fig 2C). Similarly, cell migration and invasion were reduced by silencing *DNMT3A* in vitro (Fig 2D and E). Moreover, the subcutaneous xenograft models were conducted to validate the function of *DNMT3A* in vivo. *DNMT3A* knockdown decreased tumor weight and volume (Fig 2F), which was consistent with the findings from in vitro experiments. As expected, compared with the control groups, we observed a low expression of *DNMT3A* in vivo after silencing *DNMT3A* (Fig 2G and H). Thus, we concluded that *DNMT3A*

was involved in the PC cell proliferation and metastasis, as well as tumor growth.

### DNMT3A-mediated DNA methylation regulated the expression of *SOCS2* in PC cells

According to the data from TCGA, *SOCS2* was poorly expressed in PC. As shown in Fig 3A, *SOCS2* expression was lower in PC tissues than in normal tissues. Previous studies suggested that the changes in the *SOCS2* expression might be related to the methylation modification (Fiegl et al, 2004; Xu et al, 2020). Therefore, the correlation between *SOCS2* and *DNMT3A* was analyzed. The results indicated that *SOCS2* was negatively correlated with *DNMT3A* (Fig 3B). In addition, other DNMTs, specifically *DNMT3B*, which works in a similar fashion to *DNMT3A*, were also used to examine the correlation with *SOCS2*. *DNMT3B* knockdown had no effect on *SOCS2* expression in PC cells (Fig S1A–D). We then tested the relationship between DNMT3A and *SOCS2*; the results from chromatin immunoprecipitation (ChIP) assay demonstrated that DNMT3A bound to the promoter of *SOCS2* (Fig 3C), and knockdown of *DNMT3A* induced an up-regulation of *SOCS2* at mRNA and protein levels (Fig 3D and E). *SOCS2* was reported to be a vital regulator of the JAK-STAT pathway in a variety of tumors (Letellier & Haan, 2016). Here, plasmids *SOCS2* and sh-*SOCS2* were transfected into PANC-1 and Capan-1 cells to overexpress and silence *SOCS2* expression, respectively. *SOCS2* was up-regulated after transfecting with *SOCS2*, whereas it was decreased after transfecting with sh-*SOCS2* (Fig S2A and B). Subsequently, we found that the overexpression of *SOCS2* suppressed the ratio of p-STAT5/STAT5; conversely, the knockdown of *SOCS2* expression promoted the ratio of p-STAT5/STAT5, indicating that *SOCS2* facilitated the phosphorylation level of STAT5 (Fig S2B). Interestingly, *DNMT3A* knockdown decreased the phosphorylation level of STAT5 (Fig 3F). Next, methylation-specific PCR (MSP) was performed to analyze the methylated status of *SOCS2*, as shown in Fig 3G, and *DNMT3A* knockdown resulted in an inhibitory effect on the methylation of *SOCS2*. Then, cells were treated with a demethylating agent 5-AZA to better explain the above results. We found that the 5-AZA treatment promoted the expression of *SOCS2* in PANC-1 and Capan-1 cells and inhibited the level of p-STAT5 (Fig 3H and I). These findings suggested that DNMT3A negatively regulated *SOCS2* expression by methylation modification in vitro.

### Inhibition of *SOCS2* restored the effect of *DNMT3A* knockdown in pancreatic cells

To further explore the biological function of the regulatory relationship between *DNMT3A* and *SOCS2*, we first measured the knockdown efficiency of sh-*SOCS2* in PANC-1 and Capan-1 cells. As shown in Fig 4A and B, the transfection of sh-*SOCS2* effectively reduced *SOCS2* levels and, more importantly, reversed the promoting effect of *DNMT3A* knockdown on the expression of *SOCS2*. Furthermore, we observed that the inhibition functions of *DNMT3A* knockdown on the proliferative, invasion, and migration abilities of PANC-1 and Capan-1 cells were reversed by the transfection of sh-*SOCS2* (Fig 4C–E). Therefore, *SOCS2* might exert a tumor-suppressive effect in vitro.

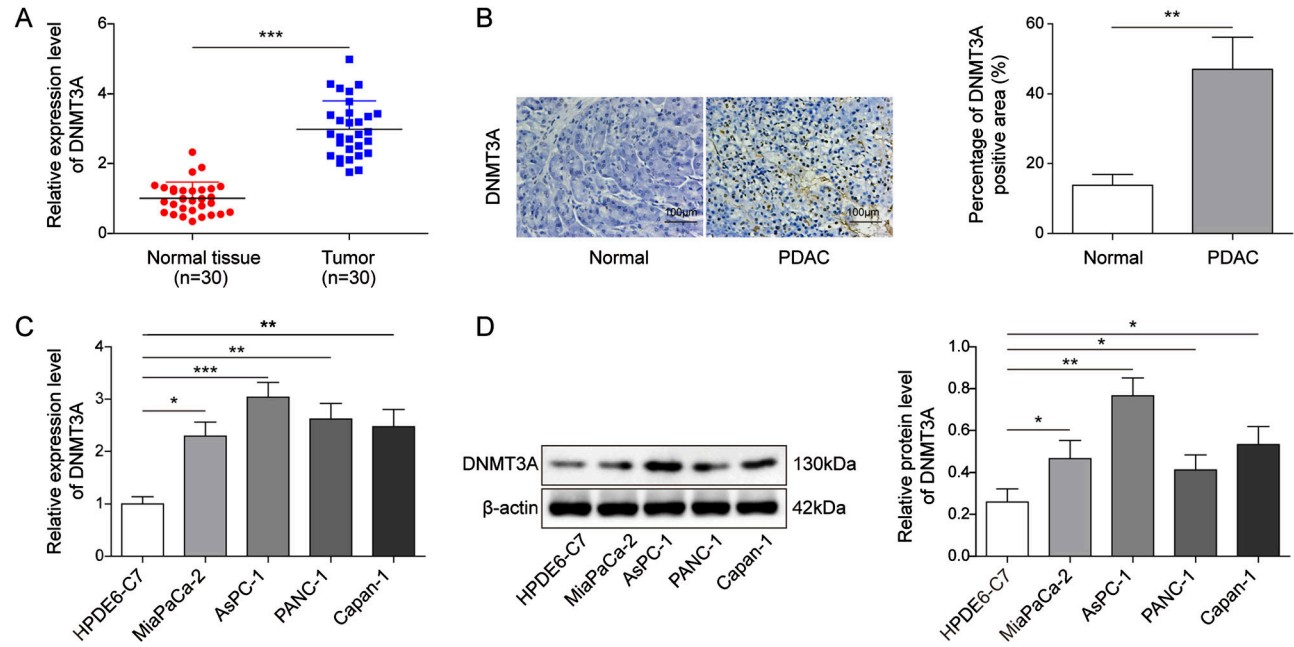

**Figure 1.** *DNMT3A* **was aberrantly up-regulated in pancreatic cancer tissues and cells.**
**(A, B)** qRT-PCR and immunohistochemistry staining suggested that *DNMT3A* was overexpressed in PC tissues compared with that of normal pancreatic tissues (n = 30). **(C, D)** A high expression of *DNMT3A* in human pancreatic duct epithelial (HPDE6-C7) cells and PC cells (MiaPaCa-2, PANC-1, AsPC-1, Capan-1) was observed by qRT-PCR and Western blot. *$P < 0.05$, **$P < 0.01$, and ***$P < 0.001$.

**Table 1.** Correlation between *DNMT3A* expression and clinicopathological characteristics in 30 cases of PC.

| Characteristics | Case (n = 30) | *DNMT3A* expression | | $\chi2$ value | *P*-value |
| | | Low (n = 17) | High (n = 13) | | |
| --- | --- | --- | --- | --- | --- |
| Age (yr) | | | | | |
| <60 | 14 | 7 | 7 | 0.475 | 0.491 |
| ≥60 | 16 | 10 | 6 | | |
| Gender | | | | | |
| Male | 15 | 6 | 9 | 3.394 | 0.0654 |
| Female | 15 | 11 | 4 | | |
| TNM stage | | | | | |
| I | 16 | 12 | 4 | 4.693 | 0.030 |
| II + III | 14 | 5 | 9 | | |
| Tumor size (cm) | | | | | |
| <4 | 17 | 12 | 5 | 3.096 | 0.079 |
| ≥4 | 13 | 5 | 8 | | |

### *miR-532-3p* targeted *DNMT3A* and regulated its expression, as well as facilitated the expression of *SOCS2* in PC cells

Previous studies suggested that miRNAs could regulate DNA methylation and other epigenetic mechanisms in neoplastic cells (Weber et al, 2007). As a widely reported tumor suppressor gene, *miR-532-3p* was indicated to be down-regulated in PC by TCGA data; this was consistent with our result (Fig 5A). In addition, we discovered that, when compared with other miRNAs with a low expression in PC and predicted to target with

*DNMT3A*, only the overexpression of *miR-532-3p* in PC cells had the most significant inhibition on *DNMT3A* (Fig S3A and B), so we focused our study on *miR-532-3p*. Considering that *DNMT3A* was highly expressed in PC tissues (Fig 1A), we examined the correlation between *DNMT3A* and *miR-532-3p*. The results showed a negative correlation between them (Fig 5B). In addition, StarBase predicted a complementary binding sequence between *DNMT3A* and *miR-532-3p* (Fig 5C). Then, the dual-luciferase reporter assay suggested that the transfection of *miR-532-3p* mimics decreased the luciferase activities of *DNMT3*-WT, whereas there was no

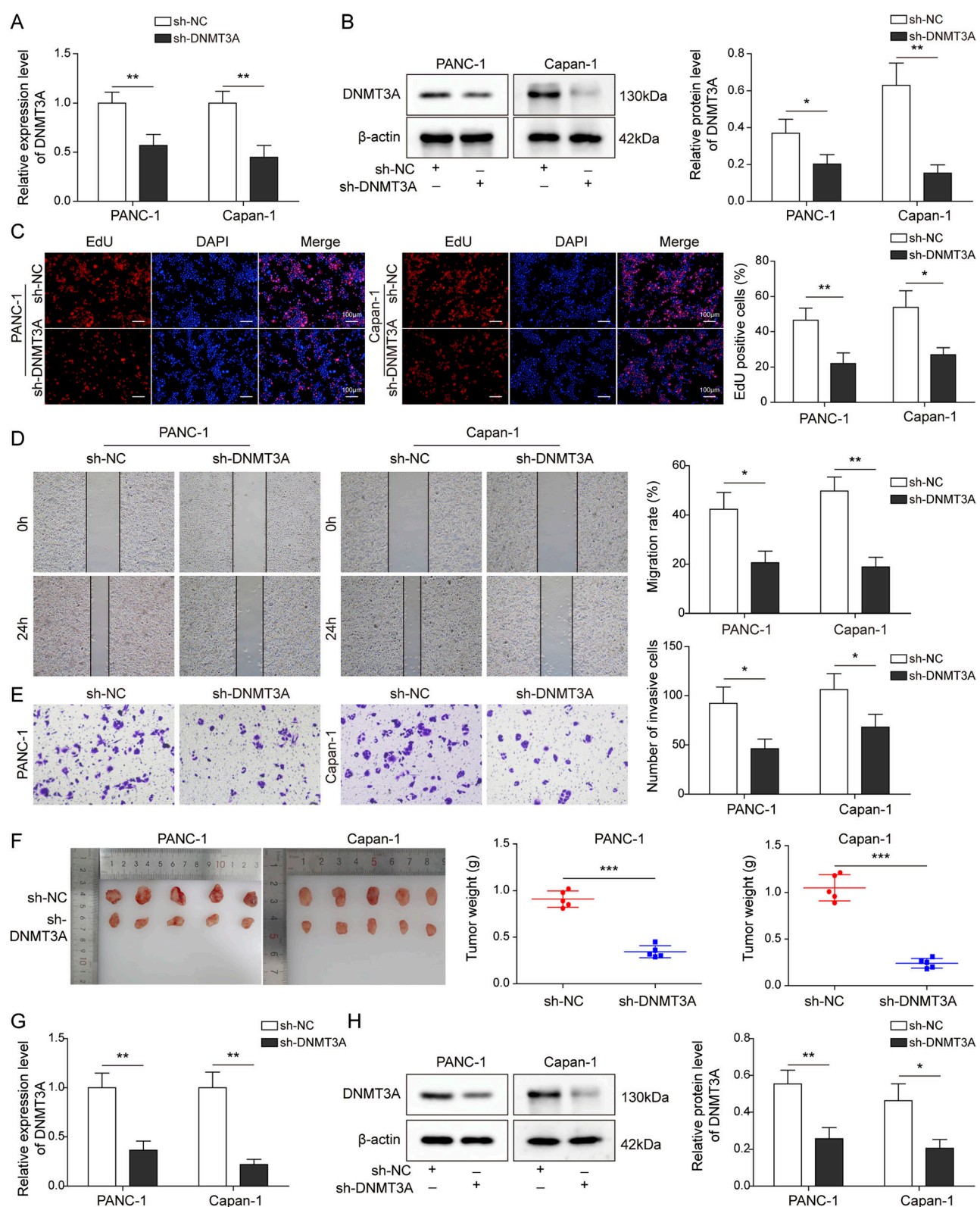

**Figure 2. DNMT3A knockdown reduced the proliferation, migration, and invasion of pancreatic cancer cells, as well as tumor formation in nude mice.**
Plasmid sh-DNMT3A was transfected into PANC-1 and Capan-1 cells for silencing DNMT3A expression. Cells were divided into two groups: sh-NC and sh-DNMT3A. **(A, B)** The knockdown efficiency was measured by qRT-PCR and Western blot. **(C)** EdU assay was performed to detect the proliferation of treated cells. **(D, E)** The migration and invasion capacities of treated cells were evaluated by wound healing and transwell assays. PANC-1 and Capan-1 cells stably expressed sh-NC or sh-DNMT3A and were

striking change after co-transfecting with *DNMT3A*-MUT and *miR-532-3p* mimics (Fig 5D). RNA immunoprecipitation (RIP) assay revealed that *DNMT3A* and *miR-532-3p* were enriched in Ago2, not in IgG, in PANC-1, and in Capan-1 cells (Fig 5E). Above all, *miR-532-3p* was directly bound to the 3′UTR of *DNMT3A* mRNA. Then, the miR-532-5p inhibitor and mimics were transfected with PC cells to silence or overexpress the expression of *miR-532-3p*. As shown in Fig 5F, miR-532-5p inhibitor and mimics were effectively transfected into PANC-1 and Capan-1 cells. We subsequently observed that the knockdown of *miR-532-3p* increased *DNMT3A* expression and STAT5 phosphorylation while decreasing *SOCS2* levels. Overexpression of *miR-532-3p* obtained the expected opposite results to miR-532-5p inhibition (Fig 5G and H). In conclusion, *miR-532-3p* regulated the expression of *DNMT3A* and *SOCS2* by targeting *DNMT3A*.

### *miR-532-3p* overexpression suppressed the progression of PC cells, as well as tumor formation in nude mice

To further investigate the function of *miR-532-3p* in the process of tumorigenesis, we performed loss- or gain-of-function experiments in vitro and in vivo. As shown in Fig 6A, the proliferative ability of PC cells was obviously decreased after the overexpression of *miR-532-3p*, whereas the knockdown of *miR-532-3p* facilitated cell proliferation. Similarly, the overexpression of *miR-532-3p* inhibited the migration and invasion capacities of PC cells (Fig 6B and C). The data from in vivo experiments suggested that the overexpression of *miR-532-3p* reduced the weight and volume of the tumor, whereas the knockdown of *miR-532-3p* resulted in an increase in tumor size (Fig 6D). Moreover, the levels of DNMT3A and p-STAT5 were down-regulated and SOCS2 was overexpressed after *miR-532-3p* mimics transfection, whereas the *miR-532-3p* inhibitor transfection obtained the opposite results (Fig 6E and F). Taken together, we speculated that the *miR-532-3p/DNMT3A/SOCS2* pathway might play an inhibitory role in the development of PC.

### *miR-532-3p* targeted *DNMT3A* and regulated PC cell progression

To explore the regulatory function between *DNMT3A* and *miR-532-3p*, the *miR-532-3p* mimics and pcDNA3.1-*DNMT3A* were co-transfected with PANC-1 and Capan-1 cells, and the results indicated that the overexpression of *DNMT3A* could reverse the inhibiting effect of *miR-532-3p* mimics on *DNMT3A* expression (Fig 7A). Subsequently, the functional experiments suggested that the exogenously expressing *DNMT3A* rescued the effect of *miR-532-3p* mimics in the proliferation, invasion, and migration of PC cells (Fig 7B–D). We therefore concluded that *miR-532-3p* impacted PC progression by targeting and negatively regulating *DNMT3A* expression.

## Discussion

Chemotherapy, radiofrequency, and surgical excision are considered the most common therapeutic approaches for PC. However, the local recurrence rates may be as high as 60% in the patients who get surgical therapy (Torre et al, 2015). The exploration of molecular targeted drugs may improve the therapeutic qualities and may achieve great progress in improving the patients' quality of life. In the present study, *miR-532-3p* was found to be down-regulated in PC and directly targeted *DNMT3A*, thereby participating in the progression of PC by regulating the expression of *DNMT3A* and *SOCS2* in vitro and in vivo.

miRNA-based therapies, such as the delivery of miRNAs to targeted cells, offer a viable therapy option for malignant tumors. *miR-532-3p* has been widely reported to show down-regulation in several types of cancer, such as renal cell carcinoma (Yamada et al, 2019) and lymphoma (Liu et al, 2020). We demonstrated the low expression of *miR-532-3p* in PC for the first time and further investigated its biological functions in the following experiments. Overexpression of *miR-532-3p* could repress the growth and metastasis of PC cells and tumor formation in nude mice, indicating a tumor-inhibiting effect of *miR-532-3p*, which is consistent with its function in other cancers. For instance, *miR-532-3p* was down-regulated and reduced the proliferative, invasion, and migration capacities of clear cell renal cell carcinoma cells by binding to TROAP (Gao et al, 2021). Jiang et al. suggested a suppressive role of miR-532-3p in prostate cancer, which was mediated by inhibiting the activation of the NF-κB pathway (Wa et al, 2020). These observations demonstrated that *miR-532-3p* could regulate gene expression in the post-transcriptional level and activate cancer-related signaling pathways through binding to downstream mRNAs. Here, *DNMT3A* was suggested to be a target gene of *miR-532-3p*.

As a well-known methyltransferase, DNMT3A has been reported to be essential for de novo methylation (Okano et al, 1999). Cytosine methylation in the CpG dinucleotide environment is an important epigenetic modification, and high levels of methylation are characteristic of transcriptional silencing on gene promoters (Robertson, 2001). Previous studies suggested that *DNMT3A* was overexpressed in various tumors, such as prostate cancer (Patra et al, 2002), breast cancer (Girault et al, 2003) and PC (He et al, 2011), which was consistent with our findings. Furthermore, we observed the promoting effect of *DNMT3A* in the progression of PC. DNMT3A-mediated hypermethylation led to the reduction of tumor suppressor genes, thereby contributing to the development of cancers (Reik et al, 2001). *DNMT3A* was a target gene of various miRNAs, such as *miR-200b* (Li et al, 2016), *miR-143* (Zhang et al, 2017), and so on. In the study of PC, *DNMT3A* knockdown restrained cell growth by mediating the inactivation of the STAT3 pathway. However, the interaction between miRNAs and *DNMT3A* in PC has not been characterized. Here, we observed a regulatory relationship between *miR-532-3p* and *DNMT3A*, and rescue assays suggested that the *miR-532-3p* overexpression promoted PC cell growth by negatively regulating *DNMT3A*.

Overexpression of *miR-532-3p* not only decreased the expression of *DNMT3A* but also up-regulated *SOCS2* levels in PC. *SOCS2* has been demonstrated to be a vital negative regulator of the JAK/STAT pathway to impact growth hormone (Croker

subcutaneously injected into BALB/c nude mice to establish xenograft mice model. **(F)** The measurement of tumor volume and weight (n = 5). **(G, H)** DNMT3A expression in mice model after silencing DNMT3A. *$P < 0.05$, **$P < 0.01$, and ***$P < 0.001$.

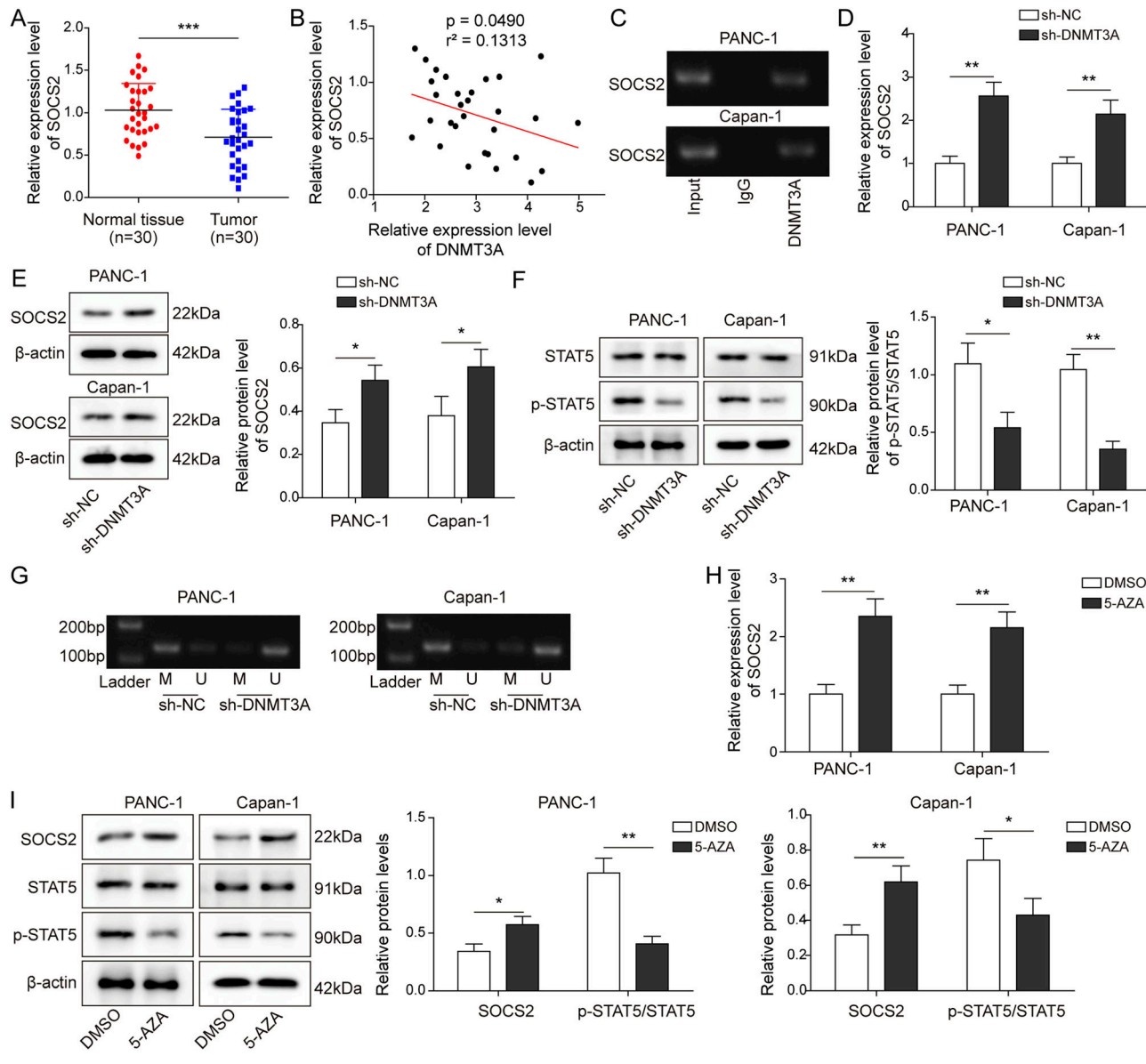

**Figure 3. DNMT3A-mediated DNA methylation regulated the expression of *SOCS2* in pancreatic cancer cells.**
**(A)** qRT-PCR analysis of *SOCS2* expression in PC and normal tissues (n = 30). **(B)** The correlation between *SOCS2* and DNMT3A. **(C)** ChIP assay of the *SOCS2* promoter region was performed with PANC-1 and Capan-1 cells by anti-DNMT3A antibody to analyze the binding sequence between DNMT3A and *SOCS2* promoter. **(D, E)** qRT-PCR and Western blot measured the expression of *SOCS2* after DNMT3A knockdown. **(F)** Western blot analysis of the phosphorylation level of STAT5, which was identified as a downstream gene of *SOCS2*. **(G)** Methylation levels in the *SOCS2* promoter were measured by methylation-specific PCR with unmethylated (U) and methylated (M) primers after DNMT3A knockdown. **(H, I)** To analyze the effect of methylation on *SOCS2* levels, demethylating agent 5-AZA was treated with PC cells. qRT-PCR and Western blot were used to detect *SOCS2* expression and the phosphorylation level of STAT5, respectively. *$P < 0.05$, **$P < 0.01$, and ***$P < 0.001$.

et al, 2008). Here, we observed that the DNMT3A-mediated DNA methylation negatively regulated *SOCS2* expression in vitro, thereby inducing an increase in the phosphorylation level of STAT5, which functions as an oncogene in various tumors (Rani & Murphy, 2016). Previous study also suggested that the overexpression of *SOCS2* could suppress STAT5 activities (Yang et al, 2012); however, *SOCS2* was epigenetically silenced through the hypermethylation of its promoter region, which was consistent with the previous finding that 5-AZA treatment

up-regulated the *SOCS2* expression in colon cancer cells (Letellier et al, 2014). This study for the first time demonstrated the related pathway and biological functions of *SOCS2* in PC and indicated that *miR-532-3p* up-regulated the expression of *SOCS2* by negatively regulating *DNMT3A*, thereby suppressing the growth and metastasis of PC cells, as well as tumor formation in nude mice.

In conclusion, *miR-532-3p* was demonstrated in this study to be a part of the *DNMT3A/SOCS2* pathway in PC. *miR-532-3p* was

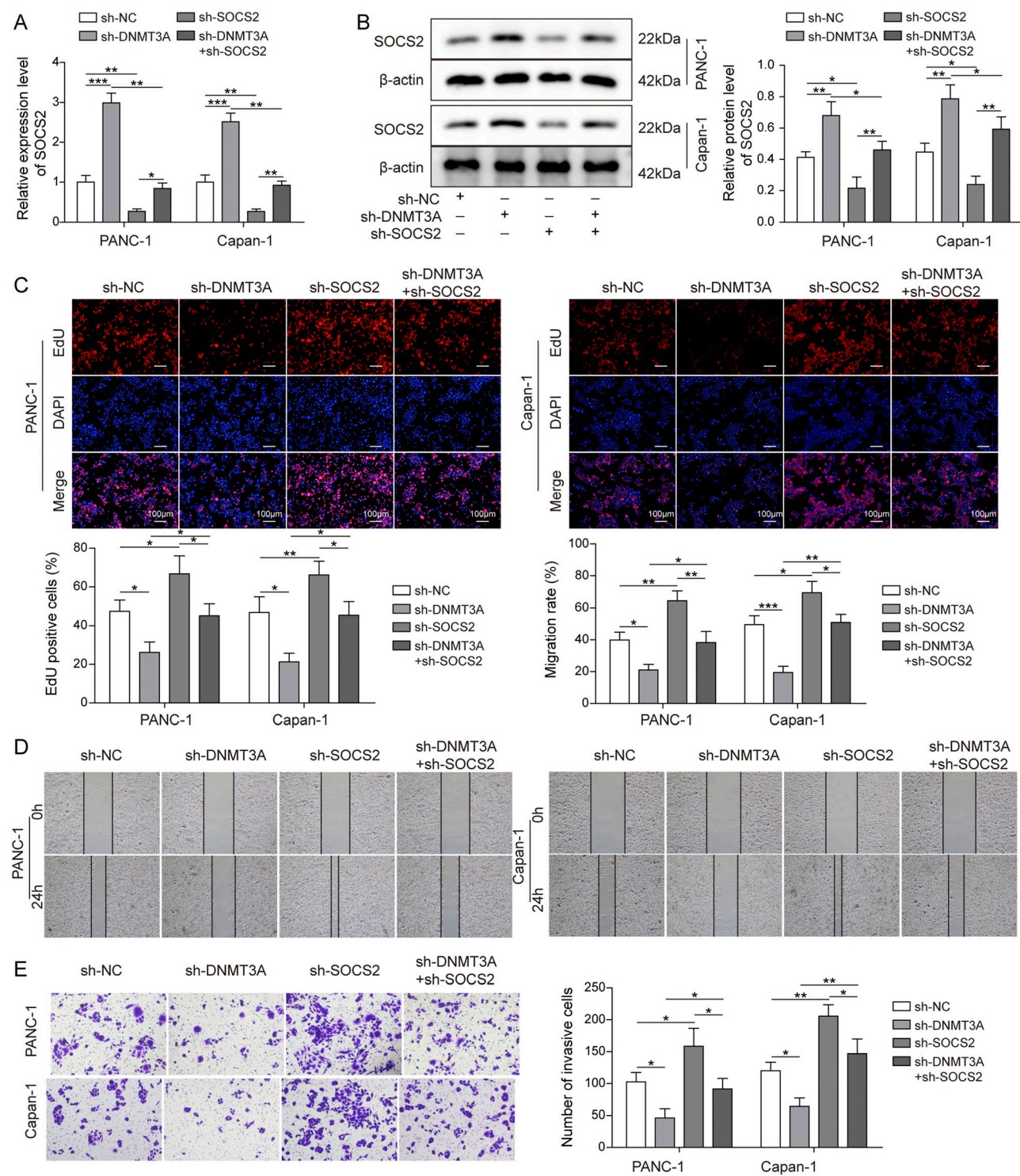

**Figure 4. Inhibition of *SOCS2* restored the effect of *DNMT3A* knockdown in pancreatic cells.**
Plasmid sh-*SOCS2* was transfected into PANC-1 and Capan-1 cells, and cells were divided into four groups: sh-NC, sh-*DNMT3A*, sh-*SOCS2*, and sh-*DNMT3A* + sh-*SOCS2*. **(A, B)** *SOCS2* expression in different groups. **(C)** EdU assay suggested the proliferation of treated cells. **(D, E)** The migration and invasion capacities of PC cells were analyzed by wound healing and transwell assays. *$P < 0.05$, **$P < 0.01$, and ***$P < 0.001$.

decreased in PC and negatively targeted *DNMT3A*. DNMT3A induced the hypermethylation of the *SOCS2* promoter and then activated STAT5 activities, thereby promoting PC tumorigenesis, whereas the overexpression of *miR-532-3p* suppressed PC progression in vitro and in vivo through the *DNMT3A*/*SOCS2* axis. These findings provided a feasible therapeutic target for PC.

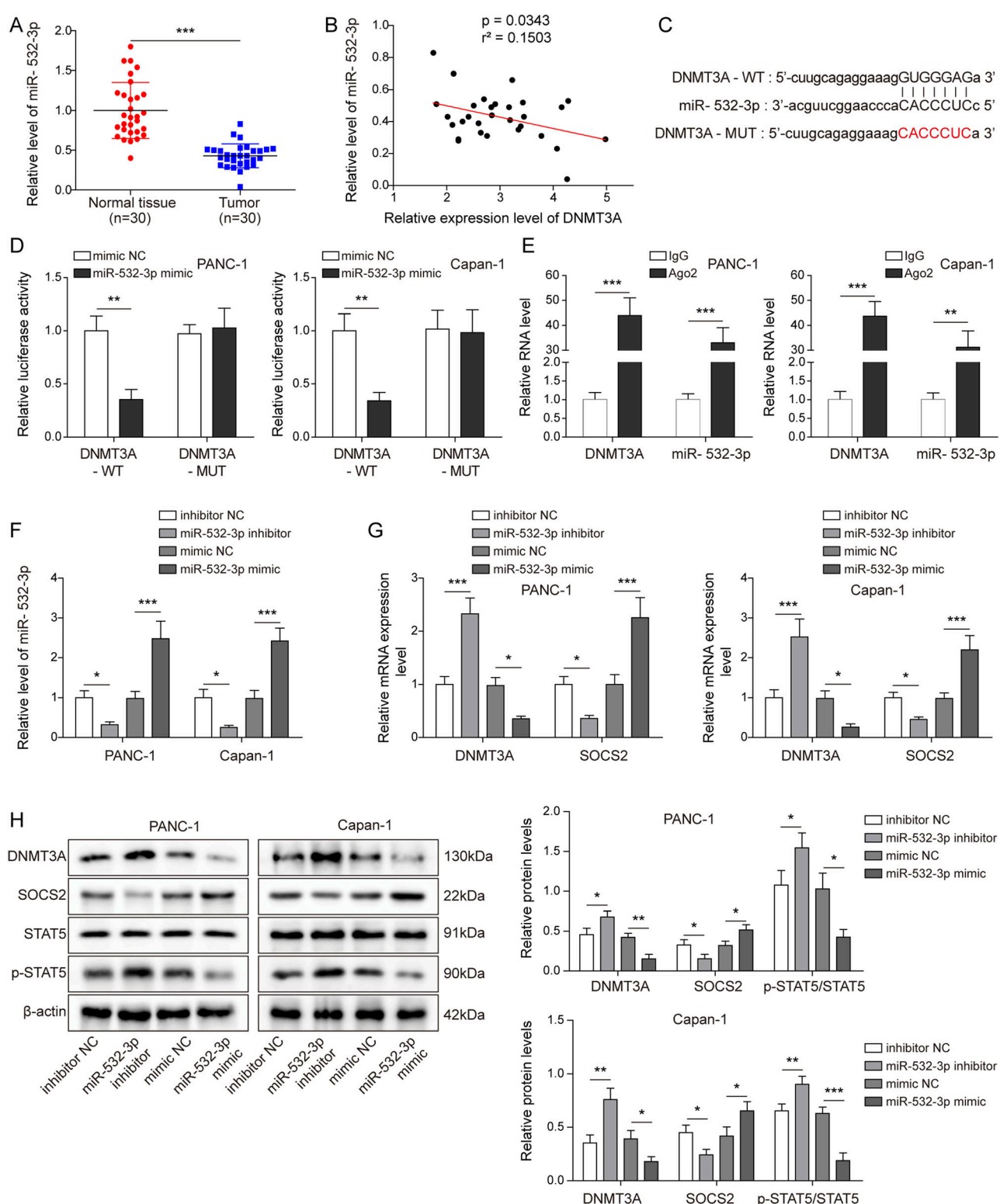

**Figure 5. *miR-532-3p* targeted *DNMT3A* and regulated its expression, as well as facilitated the expression of *SOCS2* in pancreatic cancer cells.**
**(A)** miR-532-3p was down-regulated in PC tissues. **(B)** The correlation between *miR-532-3p* and *DNMT3A*. **(C)** StarBase predicted the binding site between *miR-532-3p* and *DNMT3A*. **(D, E)** Dual-luciferase reporter and RIP assays were performed to further analyze the targeted relationship between them. *miR-532-3p* inhibitor and mimics were transfected into PANC-1 and Capan-1 cells and established four groups of cell model: inhibitor NC, *miR-532-3p* inhibitor, mimics NC and *miR-532-3p* mimics. **(F)** qRT-PCR evaluated the transfection efficiency. **(G, H)** The expression of DNMT3A and SOCS2, as well as the phosphorylation level of STAT5 were measured by qRT-PCR and/or Western blot. *$P < 0.05$, **$P < 0.01$, and ***$P < 0.001$.

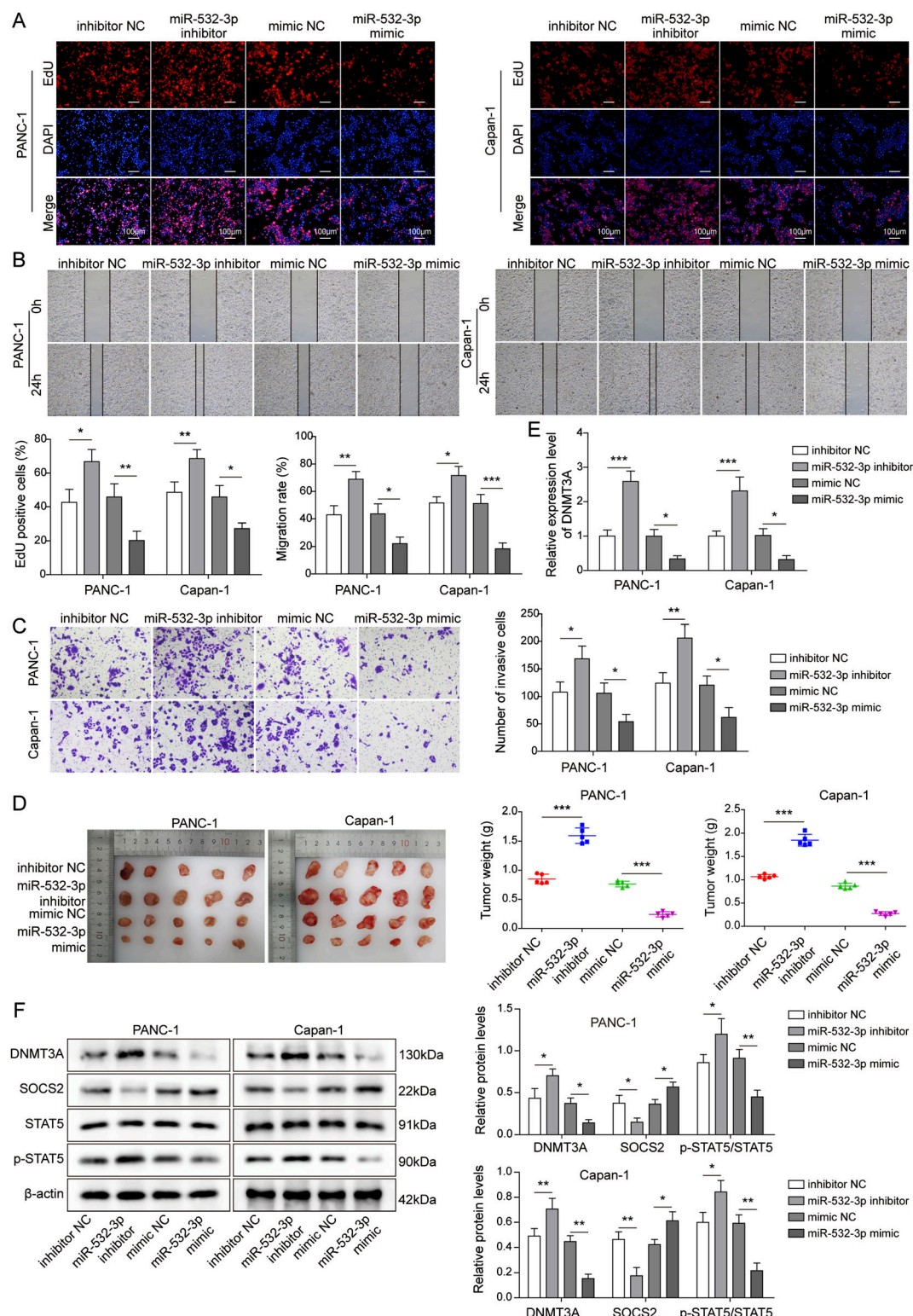

**Figure 6.** ***miR-532-3p*** **overexpression suppressed the proliferation, migration, and invasion of pancreatic cancer cells, as well as tumor formation in nude mice.**
**(A, B, C)** The effect of *miR-532-3p* knockdown or overexpression on the proliferation, migration, and invasion capacities of PC cells. **(D)** Tumor volume and weight of nude mice were evaluated after injecting with cells stably expressed *miR-532-3p* inhibitor or mimics. **(E, F)** The expression of DNMT3A and SOCS2 and the phosphorylation level of STAT5 were measured using qRT-PCR and/or Western blot in vivo. *P < 0.05, **P < 0.01, and ***P < 0.001.

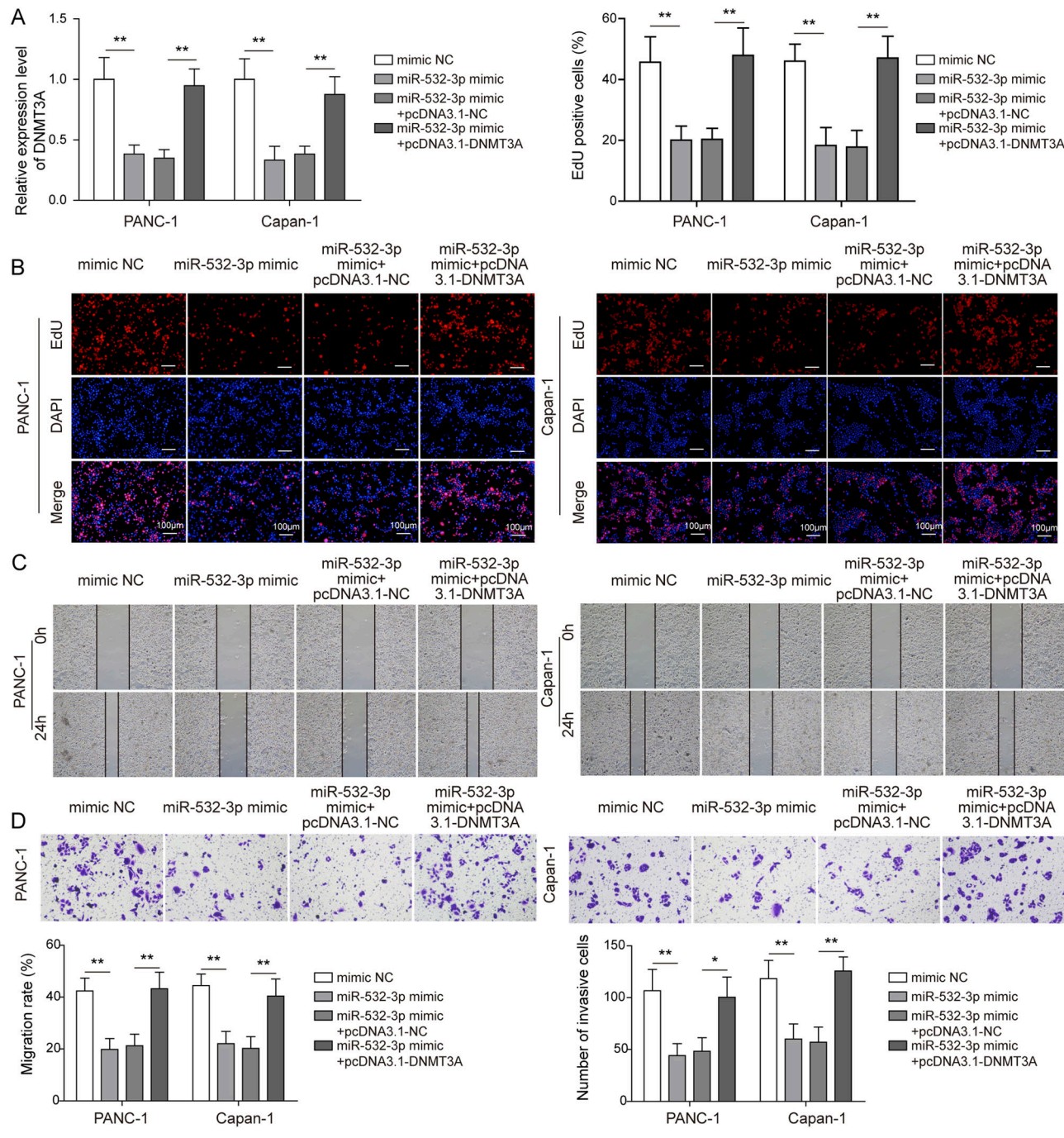

**Figure 7. miR-532-3p targeted DNMT3A and regulated pancreatic cancer cell progression.**
The DNMT3A overexpression plasmid (pcDNA3.1-DNMT3A) was used to overexpress DNMT3A in vitro. PANC-1 and Capan-1 cells were transfected with different plasmids and divided into four groups: mimic NC, miR-532-3p mimics, miR-532-3p mimics + pcDNA3.1-NC, and miR-532-3p mimics + pcDNA3.1-DNMT3A. **(A)** qRT-PCR indicated the expression of DNMT3A in different groups. **(B, C, D)** Cell proliferation, migration, and invasion capacities were analyzed. *$P < 0.05$, **$P < 0.01$, and ***$P < 0.001$.

## Materials and Methods

### Patients and clinical specimens

A total of 30 PC tissues and 30 normal adjacent non-cancer specimens were collected from patients with surgical resection at the HAINAN GENERAL HOSPITAL from Nov. 2019 to Mar. 2020. Two pathologists independently confirmed the tumor specimens. The specimens were stored at −80°C until analysis. The study was approved by The Ethics Committee of HAINAN GENERAL HOSPITAL, and all specimens were handled and anonymized according to ethical and legal standards. Written informed consents were obtained from all patients.

## Cell culture

PC cells (MiaPaCa-2, PANC-1, AsPC-1, Capan-1) and pancreatic human duct cell (HPDE6-C7) were purchased from American Type Culture Collection and then maintained in RPMI 1640 supplemented with 10% FBS (Gibco), 100 IU/ml of penicillin, and 100 μg/ml of streptomycin. All cells were grown in a humidified incubator at 37°C with 5% $CO_2$ until 80% confluency for harvesting.

## Cell transfection and treatment

The short hairpin RNAs targeting *DNMT3A* and *SOCS2* (sh-*DNMT3A*, sh-*SOCS2*) that were subcloned into pLKO.1 vector and then packaged into lentiviruses, as well as the full length of *hsa-miR-532-3p* (*miR-532-3p* mimics, agomir) and inhibitor (antagomir) were all purchased from GenePharma Company. *DNMT3A* and *SOCS2* cDNAs were amplified and inserted into pcDNA3.1 vector to obtain the overexpression plasmid: pcDNA3.1-*DNMT3A* and pcDNA3.1-*SOCS2*. Then, PANC-1 and Capan-1 cells were transfected with the lentivirus/plasmid alone or their combinations (sh-*DNMT3A* + sh-*SOCS2*, *miR-532-3p* mimics + pcDNA3.1-*DNMT3A*) using lipofectamine 3000 (Invitrogen). If required, PANC-1 and Capan-1 cells were treated with the DNMT inhibitor (5-aza-2′-deoxycytidine, 5-AZA; Sigma-Aldrich) once daily for 3 d.

## Total RNA extraction and quantitative real-time PCR (qRT-PCR)

Total RNA from tissues or cells was isolated using the TRIzol reagent (Invitrogen), and the concentration was measured by a NanoDrop Spectrophotometer (Thermo Fisher Scientific). TaqMan miRNA reverse transcription kit (Applied Biosystems) was used for miRNA qRT-PCR, with cDNA synthesized from 5 ng of total RNA. For the other genes, random primers from the RT Master Mix kit (Takara) were used to synthesize cDNAs from total RNA. The process of qRT-PCR was conducted on an ABI7500 Fast Real-Time PCR System (PE Applied Biosystems) based on the standard procedures of SYBR-Green PCR kit (Takara) following the reaction conditions: 95°C for 2 min, followed by 40 cycles of 95°C for 10 s, 60°C for 30 s. The relative expressions were normalized to that of *β-actin* mRNA or U6 using the $2^{-\Delta\Delta ct}$ method (Livak & Schmittgen, 2001). Primers are shown below. *DNMT3A*-F: 5′-CAGGAATTTGACCCTCCAAA-3′, *DNMT3A*-R: 5′-ACACCTCCGAGGCAATGTAG-3′; *SOCS2*-F: 5′-GCAAGGATAAGCGGA-CAGGT-3′, *SOCS2*-R: 5′-GTTGGTAAAGGCAGTCCCCA-3′; *hsa-miR-532-3p*-F: 5′-TCGGCAGGCCTCCCACACCCAA-3′, *hsa-miR-532-3p*-R: 5′-GTGCAGGGTCCGAGGT-3′; *β-actin*-F: 5′-TGGCACCACACCTTCTACAA-3′, *β-actin*-R: 5′-CCAGAGGCGTACAGGGATAG-3′; *U6*-F: 5′-CTCGCTTCGGCAG-CACA-3′, *U6*-R: 5′-AACGCTTCACGAATTTGCGT-3′.

## Western blot

Total proteins were extracted by RIPA buffer, and BCA protein assay kit was used to determine the corresponding concentration. Protein (25 μg) was isolated by 8% SDS–PAGE and subsequently transferred to PVDF membranes. The 1% BSA in TBS buffer was used to block the membranes, and primary antibodies were cultivated at 4°C overnight. The membrane was then washed with 1× TBST and cultivated with a secondary antibody horseradish-peroxidase-conjugated in

1×TBS at room temperature for 1 h. The membrane was washed with 1× TBST, and the protein expression was determined using SuperSignal West Pico Chemiluminescent Substrate (Pierce Biotechnology). The loading control was β-actin, detected on the same blot. All primary antibodies were purchased from Abcam: DNMT3A (ab188470; 1:2,000), DNMT3B (ab2851; 1:2,000), SOCS2 (ab109245; 1:5,000), STAT5 (ab126832; 1:1,000), P-STAT5 (ab32364; 1:1,000), and the secondary antibody (ab6802, 1:2,000).

## Immunohistochemistry

The paraffin-embedded sections were deparaffinized in xylene and rehydrated. The antigen epitopes were retrieved by microwaving in sodium citrate buffer. Diluted primary antibody DNMT3A (ab188470; 1:2,000; Abcam) was added and incubated for 24 h. After incubating with secondary antibody (ab207999, 1:2,000; Abcam), slides were stained with chromogen diaminobenzidine and then counterstained with hematoxylin. Sections were dehydrated and observed by a microscopy. The number of positive cells was counted in five random areas.

## EdU assay

The proliferative ability of treated cells was assessed using the incorporation of 5-ethynyl-29-deoxyuridine (EdU) with the EdU Assay Kit (Ribobio). Briefly, cells ($1 × 10^4$) were seeded into 96-well plates, cultured overnight, and then incubated with EdU solution (50 μM) for 2 h at 37°C. After fixing with 4% paraformaldehyde, incubating with glycine (2 mg/ml), and permeabilizing with PBS of 0.5% Triton X-100, cells were stained with Apollo reaction solution for 30 min. The cell nuclei were stained with DAPI at a concentration of 1 mg/ml for 10 min. The proportion of cells that incorporated EdU was calculated under a microscope (Nikon) in five randomly selected fields.

## Wound healing assay

Cells plated in six-well plates with $5 × 10^5$ cells/well were cultured at 37°C until 100% confluence, followed by culturing in serum-free medium for 24 h. A straight wound on the surface of the cell layer was made through using a sterile pipette. Next, debris on the cell surface was removed by washing with PBS twice, and cells were incubated in normal conditions (with 10% FBS) for 24 h. The migration of cells after 0 and 24 h of scratching was photographed with a phase-contrast microscope.

## Transwell assay

Cells harvested in serum-free medium were seeded in the upper transwell chambers (pore size, 6 μm; Corning Inc.). The bottom chamber was added with a regular medium supplemented with 10% FBS. After incubation at 37°C for 24 h, cells on the upper membrane were removed with a cotton swab, and the cells on the bottom surface of the membrane were anchored in 4% paraformaldehyde. Then, the cells were dyed at room temperature with crystal violet for 15 min. A light microscopy was employed to count the number of cells to quantify the cell invasion.

## ChIP assay

The kit from Millipore was used to perform ChIP assay. Briefly, cells were cross-linked with 1% formaldehyde for 10 min and then lysed and sonicated to obtain chromatin fragments of 500 average size. 1% of supernatant was collected to serve as an input control. The chromatin diluted by ChIP solution was immunoprecipitated with DNMT3A antibody or IgG at 4°C overnight with rotation. After reversing the cross-links, the immune complexes were purified and analyzed by qRT-PCR.

## Methylation-specific PCR

To measure the methylation status of *SOCS2*, genomic DNAs were extracted by an extraction kit (Tiangen). After modifying with bisulfite, genomic DNAs were purified, and qRT-PCR analysis was performed, followed by agarose gel electrophoresis and visualization by a gel imaging system.

## Bioinformatics and dual-luciferase reporter assay

The binding sites between *miR-532-3p* and *DNMT3A* were predicted by StarBase (http://mirwalk.umm.uni-heidelberg.de). QuikChange Mutagenesis kit was then used to generate the mutations. 3′-UTR sequences of *DNMT3A* containing WT or mutated binding site were subcloned into pRL-TK luciferase reporter. All constructs were sequenced to verify integrity. The PANC-1 and Capan-1 cells were transfected with 300 ng of firefly luciferase reporter and 25 ng of Renilla luciferase plasmid plus 900 ng of empty vector or miRNA mimics. After 24 h of transfection, luciferase assays were done using Dual Luciferase Reporter Assay kit (Promega), and the ratio of Firefly to Renilla luciferase activity was determined.

## RIP assay

RIP assay was conducted using a Magna RIPTM RNA-Binding Protein Immunoprecipitation Kit (Millipore). Cells were lysed with RIP lysis buffer and then immunoglobulin G antibody (anti-IgG) and argonaute 2 antibody (anti-Ago2) coated on magnetic beads overnight. Then, the magnetic bead-bound complexes were immobilized with a magnet and unbound materials were washed off. Part of the cell was used as the negative control, named input. The co-precipitated RNA was extracted using TRIzolTM, and qRT-PCR was then used to analyze the purified RNA.

## Tumor xenograft assay

BALB/c-nude mice (20–22 g, 4 wk) were purchased from Animal Experiment Center of the Chinese Academy of Sciences (Shanghai, China) and used as xenograft animal models. Animals were raised in specific pathogen-free conditions, and all the animal experiments were approved by the animal ethics committee of HAINAN GENERAL HOSPITAL. The mice were randomly divided into experiment groups, PANC-1 and Capan-1 cells ($1 \times 10^7$) stably expressing sh-*DNMT3A* were injected subcutaneously into the right flank of the nude mice under aseptic conditions, and the *miR-532-3p* inhibitor (antagomir) and *miR-532-3p* mimics (agomir) were injected into the mice via the tail vein for three consecutive days (80 mg/kg/d) Then, the tumor size was measured by calipers every 4 d, and after 28 d, mice were euthanized, and then xenograft tumor tissues were harvested and stored at a –80°C refrigerator for analyzing mRNA and protein levels.

## Statistical analysis

Data were given as the mean and SD. *t* test was used to compare the difference between two groups for continuous variables. One-way ANOVA followed by the Tukey's post hoc test was used for multiple comparisons. All the analyses were performed using GraphPad Prism 6 (GraphPad Software, Inc.). $P < 0.05$ was considered statistically significant.

# Data Availability

The datasets used or analyzed during the current study are available from the corresponding author on reasonable request.

# Supplementary Information

# Acknowledgements

This work was supported by Key Research and Development Project of Hainan Province (Social Development) (No.ZDYF2022SHFZ131): Study on the mechanism of Mtif2-induced mitochondrial metabolic reprogramming inhibiting AIFM1 and promoting 5-FU resistance in liver cancer.

## Author Contributions

K Wang: conceptualization, data curation, formal analysis, validation, investigation, visualization, methodology, and writing—original draft.
D Gong: conceptualization, data curation, formal analysis, validation, investigation, visualization, methodology, and writing—original draft.
X Qiao: resources and software.
J Zheng: conceptualization, supervision, funding acquisition, project administration, and writing—original draft, review, and editing.

## Conflict of Interest Statement

The authors declare that they have no conflict of interest.

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
