## [Reviewer comments · Life Science Alliance]

MiR-532-3p inhibited the methylation of SOCS2 to suppress the progression of PC by targeting DNMT3A

Kaiqiong Wang, Dongwei Gong, Xin Qiao and Jinfang Zheng

DOI: 10.26508/lsa.202201703

Corresponding author(s): Dr. Jinfang Zheng (Hainan General Hospital)

Review timeline:

Submission Date:	2022-09-02
Editorial Decision:	2022-10-24
Revision Received:	2022-12-27
Editorial Decision:	2023-01-06
Revision Received:	2023-01-13
Accepted:	2022-01-13

Scientific Editor: Novella Guidi

Transaction Report:

No Peer Review Process File is available with this article, as the authors have chosen not to make the review process public in this case.

Dr. Jinfang Zheng
Hainan General Hospital
NO.19, Xiuhua Road
Haikou 570311
China

Dear Dr. Zheng,

Thank you for submitting your manuscript entitled "MiR-532-3p inhibited the methylation of SOCS2 to suppress the progression of PC by targeting DNMT3A" to Life Science Alliance. The manuscript was assessed by expert reviewers, whose comments are appended to this letter. We invite you to submit a revised manuscript addressing the Reviewer comments.

Thank you for this interesting contribution to Life Science Alliance. We are looking forward to receiving your revised manuscript.

Sincerely,

-- High-resolution figure, supplementary figure and video files uploaded as individual files: See our detailed guidelines for preparing your production-ready images,

<https://www.life-science-alliance.org/authors>

B. MANUSCRIPT ORGANIZATION AND FORMATTING:

2nd Editorial Decision

06 January 2023

RE: Life Science Alliance Manuscript #LSA-2022-01703R

Dr. Jinfang Zheng
Hainan General Hospital
NO.19, Xiuhua Road
Haikou 570311
China

Dear Dr. Zheng,

Thank you for submitting your revised manuscript entitled "MiR-532-3p inhibited the methylation of SOCS2 to suppress the progression of PC by targeting DNMT3A". We would be happy to publish your paper in Life Science Alliance pending final revisions necessary to meet our formatting guidelines.

- please add ORCID ID for the corresponding author-you should have received instructions on how to do so
- please add a Summary Blurb/Alternate Abstract to our system
- please add the Twitter handle of your host institute/organization as well as your own or/and one of the authors in our system
- please add your supplementary figure legends to the main manuscript text after the references section. All figure legends should only appear in the main manuscript file
- please add an Author Contributions section to your main manuscript text
- please consult our manuscript preparation guidelines <https://www.life-science-alliance.org/manuscript-prep> and make sure your manuscript sections are in the correct order
- please add a callouts for Figures S2A-D and S3A, B to your main manuscript text
- figures should be in PowerPoint, TIFF, PDF, or EPS format

FIGURE CHECKS:

- please add sizes next to all blots
- all figures need to be on a single page (Figure 6)

A. FINAL FILES:

B. MANUSCRIPT ORGANIZATION AND FORMATTING:

Sincerely,

RE: Life Science Alliance Manuscript #LSA-2022-01703RR

Dr. Jinfang Zheng
Hainan General Hospital
NO.19, Xiuhua Road
Haikou 570311
China

Dear Dr. Zheng,

Thank you for submitting your Research Article entitled "MiR-532-3p inhibited the methylation of SOCS2 to suppress the progression of PC by targeting DNMT3A". It is a pleasure to let you know that your manuscript is now accepted for publication in Life Science Alliance. Congratulations on this interesting work.

DISTRIBUTION OF MATERIALS:

Again, congratulations on a very nice paper. I hope you found the review process to be constructive and are pleased with how the manuscript was handled editorially. We look forward to future exciting submissions from your lab.

Sincerely,
